# Computational Portable Microscopes for Point-of-Care-Test and Tele-Diagnosis

**DOI:** 10.3390/cells11223670

**Published:** 2022-11-18

**Authors:** Yinxu Bian, Tao Xing, Kerong Jiao, Qingqing Kong, Jiaxiong Wang, Xiaofei Yang, Shenmin Yang, Yannan Jiang, Renbing Shen, Hua Shen, Cuifang Kuang

**Affiliations:** 1School of Electronic and Optical Engineering, Nanjing University of Science and Technology, Nanjing 210094, China; 2The Affiliated Suzhou Hospital of Nanjing Medical University, Suzhou Municipal Hospital, Suzhou 215002, China; 3School of Optoelectronic Science and Engineering & Collaborative Innovation Center of Suzhou Nano Science and Technology, Soochow University, Suzhou 215006, China; 4College of Optical Science and Engineering, Zhejiang University, Hangzhou 310027, China; 5Research Center for Intelligent Sensing, Zhejiang Lab, Hangzhou 311100, China

**Keywords:** computational imaging, portable microscope, point-of-care-test

## Abstract

In bio-medical mobile workstations, e.g., the prevention of epidemic viruses/bacteria, outdoor field medical treatment and bio-chemical pollution monitoring, the conventional bench-top microscopic imaging equipment is limited. The comprehensive multi-mode (bright/dark field imaging, fluorescence excitation imaging, polarized light imaging, and differential interference microscopy imaging, etc.) biomedical microscopy imaging systems are generally large in size and expensive. They also require professional operation, which means high labor-cost, money-cost and time-cost. These characteristics prevent them from being applied in bio-medical mobile workstations. The bio-medical mobile workstations need microscopy systems which are inexpensive and able to handle fast, timely and large-scale deployment. The development of lightweight, low-cost and portable microscopic imaging devices can meet these demands. Presently, for the increasing needs of point-of-care-test and tele-diagnosis, high-performance computational portable microscopes are widely developed. Bluetooth modules, WLAN modules and 3G/4G/5G modules generally feature very small sizes and low prices. And industrial imaging lens, microscopy objective lens, and CMOS/CCD photoelectric image sensors are also available in small sizes and at low prices. Here we review and discuss these typical computational, portable and low-cost microscopes by refined specifications and schematics, from the aspect of optics, electronic, algorithms principle and typical bio-medical applications.

## 1. Introduction

Disease cross-regional transmission (e.g., COVID-19 explosion), food safety (e.g., pathogenic Escherichia coli) and environmental pollution (e.g., water eutrophication) might break out suddenly and unexpectedly [1,2,3,4,5,6]. These have threatened human health, life and the environment. Biological and medical detection and analysis are the important ways to prevent and control these problems. For biological and medical detection and analysis, a microscope is a very important optical instrument, one which directly promotes the development of biotechnology, medicine and pathology. Nowadays, research-level and desktop microscopes have been widely used in medical centers and scientific institutes [7,8,9,10,11,12,13,14,15,16,17,18], including bright-field microscopes, dark-field microscopes, fluorescence microscopes, Zernike-phase-contrast (ZPC) microscopes, differential-interference-contrast (DIC) microscopes, laser confocal microscopes and super-resolution microscopy imaging systems, etc. These traditional microscopy imaging systems are suitable for hospitals and research-level laboratories. They are expensive and bulky. And their operations and analysis require professional skill. Additionally, the medical centers and scientific institutes might be far away from the outdoor biological/medical sampling sites. The above issues all affect detection speed and efficiency. Especially in some underdeveloped remote areas, the medical conditions and resources cannot guarantee effective detection and analysis. Further, with the development of mobile/online medicine, daily medical inspection demands arise quickly. Addressing the above demands, point-of-care-test (POCT) devices and techniques attract many researchers and engineers for observation and tele-diagnosis [19,20,21,22,23,24,25,26].

The POCT aims to quickly collect samples, analyze results, and provide test reports. Small and simplified-operation POCT microscopes are particularly desirable [19,20,21]. POCT diagnostic microscopy devices generally need some of the following characteristics: (1) results are timely: the test results can be quickly provided at the site, without the need to wait for several days to print a report; (2) the operating system is simple: anyone can operate the instrument without special training, and it can be used in a wide geographical range without regional restrictions; (3) accurate qualitatively or quantitatively: instant detection not only requires rapid detection results, but also relatively accurate results; qualitative and quantitative accuracy is the most critical means to achieve user promotion, and POCT’s detection should rival the results of professional instrumentation equipment; (4) portability: to achieve rapid detection in any location, small and portable is an essential factor.

With the development of consumer-level electronic and optical products, modern opto-electronic elements (e.g., laser diodes (LDs) or light-emitting diodes (LEDs), optical fibers and binary optical components, CMOS/CCD photoelectric image sensors) are characterized by smaller sizes and lower prices. These promote portable biomedical testing platforms, mainly focusing on spectroscopy [22], microscopy and so on. Distinct from spectroscopy, POCT microscopy provides direct 2D-vision information. POCT microscopy devices are becoming cheaper and smaller. More importantly, we are in the 5G+ telecommunication and AI era [27,28]. High-performance processing chips are widely used in daily life. For example, the advanced central processing unit (CPU) and graphic processing unit (GPU) enable excellent computing hash-rate and image processing capabilities in a laptop. The Bluetooth module, WLAN module and 3G/4G/5G module are sufficiently miniature and low-priced. The pinhole lens, industrial monitor lens and microscopy objective lens are also characterized by small size and low price. As a typical example, smartphones simultaneously take into account good optical imaging performance, advanced image sensing equipment, high-performance processing chips and big-data telecommunication, functionalities which can be easily achieved by the industry. The coming 5G+ telecommunication era will quicken big-data transmission. Data interaction and processing can be carried out between the machine with wireless access to the remote super-computing server workstation and cloud storage space. Given these conditions, computational, portable and low-cost microscopes evolve quickly, serving for POCT and tele-diagnosis [19,20,21,27,28]. Here we would review these typical cases as a sort of imaging theorem, mainly including lens-free microscopes, smart-phone microscopes, singlet microscopes and portable super-resolution microscopes.

## 2. Lens-Less Microscopy

### 2.1. Projection Lens-Less Microscopy

Projection imaging is the simplest and earliest lens-less microscopy method. The entire process does not require unique image reconstruction algorithms [29,30,31,32,33,34,35,36,37,38,39]. Figure 1 is a typical schematic of lens-less diffraction microscopy setups based on a single frame [39], in projection lens-less microscopy, the sample is placed directly on the CCD/CMOS image sensor. A light source, e.g., a LED, directly illuminates the sample, and the projected shadow of the sample is directly captured by the CCD/CMOS image sensor. Assuming the distance from the light source to the sample is z1, the gap from the sample to the image sensor is z2. The light modulated by the sample information propagates along z2 to the CMOS image sensor. When the light source is incoherent, the directly recorded image is a defocused blur image at the out-of-focus distance of z2. When the light source is (partially) coherent, the recorded image may be the diffracted pattern with concentric interference fringes. These defocused blurs and concentric interference fringes both deteriorate the image’s resolution abilities and the image’s contrast. When the image is incoherent, the blur might be optimized by the computational deconvolution algorithms, such as the famous Richardson-Lucy method. When the image is diffracted, the resolution might be improved by the hologram reconstruction algorithms. Besides, other hardware processes, such as placing a hole-array between the sample and the image sensor, might be considered to improve the resolution. An opaque metal layer is coated in the image sensor, and an array of sub-micron small holes are arranged on the metal layer, where each hole corresponds to each pixel on the image sensor. This projection lens-less microscopy method effectively reduces the volume of the imaging system while improving the imaging resolution. However, put briefly, the projection lens-less microscopy is mainly limited by its resolution ability.

### 2.2. Fluorescence Lens-Less Microscopy

Fluorescent labelling methods are widely used in bio-medical and pathological observing. The CMOS image sensor records the emitted fluorescent light, not the light source’s rays. Usually, the wavelength spectrum of the emitted light and that of the light source are different. It is generally necessary to place a filter between the sample and the CMOS image sensor to filter out the original wavelength spectrum of the light source in a fluorescent lens-less microscope [40,41,42,43,44,45,46]. However, even with a wavelength spectrum filter, there may still be some leakage of the light source’s rays into the CMOS image sensor. Other methods are used to reduce these leaked rays, such as adding a total internal reflection (TIR) prism. The resolution ability of the lens-less fluorescent microscopy is limited by the system’s point spread function (PSF). The PSF can be tested and calculated by recording light from a single point in the sample plane. The light would spread to a spot on the CMOS image sensor. Then, the PSF size would be measured. Therefore, it can be seen that the sample-to-sensor distance will affect the PSF. For example, when the defocusing gap is ~200 μm, the PSF limits the resolution to ~200 μm, which is far worse than the resolution of most lens-based microscope systems. Therefore, despite the obvious advantages of lens-less fluorescence imaging devices in terms of cost, portability and field-of-view, the spatial bandwidth product obtained by this method is not superior to conventional bench-top microscopes. In order to improve the resolution of lens-less fluorescent microscopy, there are several methods: (1) Place the filter directly on the sensor, so that defocusing gap can be minimized. Using this method, the spatial resolution can be improved to ~10 μm. (2) The fluorescent light is relayed using a densely packed fiber array. In principle, when these fibers deliver light to the image sensor, the fiber beam is amplified, providing a degree of magnification for lens-less imaging. Using this method, the resolution can be improved to ~4 μm. (3) The mask of the nanostructure would be placed close to the object; the PSF of the system is no longer spatially invariant, but will depend on nanostructures. With this approach, sub-pixel resolution of up to 2–4 μm can be achieved. (4) computational deconvolution and compression decoding algorithms could be as the post process.

### 2.3. Digital Holographic Lens-Less Microscopy

When the light source is spatially and temporally coherent, the diffracted pattern recorded by the CMOS image sensor are the on-axial holograms. Different from projection lens-less microscopy and fluorescent lens-less microscopy, digital holographic lens-less microscopy uses the recorded holograms to computationally reconstruct the intensity and phase information of the sample [47,48]. In general, lens-less microscopy based on digital holographic reconstruction is essentially on-axial digital holographic reconstruction. In principle, the acquired image is the light $U_{O}\left(x,y\right)$ scattered by the object on the sample and the the reference light $U_{R}\left(x,y\right)$ of the transparent substrate:(1)$$U\left(x,y\right)=U_{R}\left(x,y\right)+U_{o}\left(x,y\right)=A_{R}+A_{O}\left(x,y\right)\exp\left[i\varphi_{o}\left(x,y\right)\right],\;$$
where $A_{R}$ are the amplitude information of the reference light and $A_{O}$ is the object light, respectively; $\varphi_{o}$ is the phase information of the object light. The digital holographic reconstruction seeks to reconstruct the object light through the directly recorded hologram I(x,y). The transfer function of coherent wave propagation is
(2)Hz(x,y)={exp(ikz1−λ2fx2−λ2fy2), fx2+fy2≤1λ20, fx2+fy2≥1λ2 , 
where λ is the wavelength of the illumination wavelength. *f*_x_ and *f*_y_ are the spatial frequency domain coordinates.

#### 2.3.1. Phase Recovery Based on Single Frame

In lens-less holographic microscopy, the conjugate image is unavoidable. The simplest single-frame phase recovery method is an iterative phase recovery method based on support domain constraints [49,50,51,52,53]. The key to this approach is the support field/mask of the object plane. For example, one can use a simple threshold or segmentation algorithm to automatically estimate the position of the object, computationally creating an object-dependent support field/mask. Also, a hardware mask can be created with Talbot grating illumination. The collected single-frame object image is used as the amplitude information of the complex amplitude $U_{2}^{O}$ of the initialization object, and an arbitrary value is used as the phase. The complex amplitude $U_{2}^{k+1}$ (denoted as $U_{2}^{O}$ for the first time, and k is the number of times the complex amplitude is limited to be updated) is propagated to the object plane using the angular spectrum propagation, and the light intensity information propagated to the plane is updated by using the predicted support domain information, and the phase is kept unchanged, a new complex amplitude $U_{1}^{k+1}$ is obtained, and the complex amplitude is propagated back to the image plane. Similarly, the light intensity information of the complex amplitude $U_{2}^{k}$ at this time is updated by using the hologram of the directly collected image plane to obtain a new $U_{2}^{k+1}$, then the loop iterates to get the final complex amplitude information. This single-frame-based lens-less microscopic imaging method is generally simple in structure, so the system is compact. Thus, it would be widely used in places with low requirements for imaging quality and high requirements for portability, such as water pollution monitoring. In addition, with some additional devices (which may slightly increase the volume of the system), cell movement monitoring can be achieved.

#### 2.3.2. Phase Recovery Based on Multiple Holograms

Multiple-hologram phase recovery methods can be classified into those based on multi-distance recorded holograms, multi-wavelength illumination, and multi-angle illumination [39,54,55,56,57,58]. The typical image reconstruction is essentially the Gerchberg-Saxton (G-S) iterative phase recovery method. The typical experimental setup is shown in Figure 2; the light source directly illuminates the sample, the sample is axially scanned along the *z*-axis, and holograms are directly recorded by the CMOS image sensor at different distance planes. For computational reconstruction, firstly the holograms collected in each plane should be registered. Then the image at the first plane (phase initialized to a zero 2D-matrix) is propagated to the second plane. The phase remains unchanged, and the amplitude is substituted by the square root of the intensity at the second plane. A new complex amplitude is synthesized, and then is afterwards propagated computationally. These same steps are executed to the n-th plane, completing one iteration loop. After completing multiple iterations, the reconstructed complex amplitude is computationally propagated to the focal plane. The high-resolution complex amplitude information (i.e., amplitude and phase) of the object would be obtained, where there is no twin-image-noise.

The above multi-distance-based phase recovery method needs to collect intensity holograms at different defocus planes, so a mechanical displacement device is generally required. To avoid this problem, Zuo et al. proposed a method based on multi-wavelength illumination to achieve the phase recovery of objects. The whole reconstruction processes are as follows: holograms under different wavelengths are collected, such as R/G/B illumination. Then, the TIE solver is used to guess the initial value, to obtain the complex amplitude of the object under G light illumination. According to the diffraction integral equation, the wavelength λ and the propagation distance z always appear in pairs. The change of the wavelength is equivalent with the change of the propagation distance. Therefore, for a non-scattering sample, if the illumination wavelength is changed from λ to λ + Δλ, the collected diffraction pattern can be equivalently seen as a change in propagation distance Δz=z·Δλ/λ. Holograms of different wavelengths can be regarded as images collected at different defocus planes under the same illumination wavelength. Finally, the high-resolution complex information would be obtained with no twin-image-noise, using the G-S iterative method.

### 2.4. Deep Learning Lens-Less Microscopy

The deep learning lens-less holographic phase recovery algorithm is without iterative optimization processes [59,60,61,62,63]. The deep learning algorithm to reconstruct the single-frame complex information by convolution neural network (CNN) architecture is shown in Figure 3. The training set is obtained, and the trained neural network can recover the corresponding complex amplitude of the object according to the input diffraction image.

### 2.5. Colorful Lens-Less Microscopy

The high-resolution lens-less microscopy relies on the hologram recorded on the CMOS image sensor. When the illumination source is completely incoherent, the resolution will be relatively poor. Therefore, in order to improve the coherence of the illumination light source, quasi-monochromatic light is generally selected for illumination, which is within a narrow spectrum band. However, in many bio-medical observing cases, valuable information on biological samples is generally stained by chemical dying. Bio-medical researchers prefer to use color images for diagnosis, such as the H&E staining. Therefore, researchers have proposed lens-free color imaging [63,64,65,66]. One of the simplest methods is statistical based color mapping. Greenbaum et al. used this method to achieve color imaging of human breast cancer, Pap smear, etc. [63]. Another relatively simple colorization method is to firstly illuminate the object with monochromatic red, green, and blue light sources, acquiring three images in sequence. Then, by superposing the reconstruction results at the three illumination wavelengths, the final color reconstructed image is obtained. Additionally, the deep learning’s virtual colorizing would also achieve colorful lens-less microscopy under only one coherent illumination, as shown in Figure 4 and Figure 5.

### 2.6. 3D Tomographic Lens-Less Microscopy

In the field of biological research, fast and compact 3D structure imaging and analysis are very attractive. The schematic diagram of 3D tomographic lens-less microscopy using at least two angular illuminations is shown in Reference [67]. In the lens-less 3D tomography based on multi-angle illumination, the height of the micro-object is determined by calculating the lateral displacement of the hologram [67,68,69,70,71]. This 3D tomography is based on the Fourier diffraction projection theory. For example, the multi-angle and multi-wavelength illumination are provided by the LED array to realize the 3D tomographic microscopy [67]. Using such techniques, a spatial resolution of 1 μm × 1 μm × 3 μm is achieved in the x, y, and z directions within a large volume of approximately 15 mm. The sample is placed directly on the CMOS image sensor, and a partially coherent light source is ~70 mm away from the CMOS image sensor, and the sample is illuminated from different angles (±50°) to obtain lens-less on-axial holograms at different angles.

### 2.7. Lens-Less Microscopy Application Examples for POCT and Biomedicine Diagnosis

Projection lens-less microscopy. Zheng et al. developed a projection lens-less microscope to track cell growth and differentiation of embryonic stem cells [29]. This imaging platform enables high-resolution and autonomous imaging of cells plated or grown on low-cost CMOS sensor chips, with the ability to image samples with a FOV with 6 mm × 4 mm with a resolution of 660 nm. The total data acquisition process takes about 20 s, and the reconstruction time of a single digital image is less than 1 s. Their cell culture experiments showed that their application of this technique could be a useful tool for long-term cell observation in vitro. At the same time, their device mainly consists of a smartphone and Lego bricks, demonstrating that their imaging platform can be easily assembled.

Fluorescence lens-less microscopy. Several researchers have successfully imaged fluorescent C. elegans samples on a fluorescence lens-less microscopy platform enhanced by compression encoding algorithms [42]. The imaging FOV is >8 cm^2^. The spatial resolution is about 10 µm. Additionally, they tested the effectiveness of this on-chip imaging approach using different types of imaging sensors, which achieves similar resolution ability independent of the imaging sensor chip. This provides a useful tool for high-throughput screening applications in observing fluorescent biomedical samples.

Digital holographic lens-less microscopy. Although there are many reconstruction methods for the digital holographic lens-less microscopy, the application examples are almost similar to each other. The resolution abilities, FOV and the imaging speed are similar to each other. For example, Cedric Allier et al. provide a method based on holographic lens-less video microscopy to measure important metrics for cell proliferation studies, e.g., cell cycle-duration and cell dry mass [35]. The microscopy imaging FOV is ~29.4 mm^2^, and the resolution ability of the reconstructed phase image is ~5 µm. By tracking 2.7 days’ acquisition of HeLa cells in culture, they would get a dataset with 2.2 × 106 single-cell morphology measurements and 10,584 cell cycle trajectories, with the speed of 1/10 frame per minute across the FOV. In addition to this study on cell proliferation, the setup could be used for the study of cell migration by tracking a large number of trajectories simultaneously over several hours. Deep learning computational methods, coloring and 3D tomography would enhance the holographic lens-less microscopy. For example, Liu et al. demonstrated a deep neural network-based virtual staining technique for label-free cells, named PhaseStain [62]. It is based on the quantitative phase image (QPI) obtained by lens-less microscopy and converted into a stained image equivalent to a bright-field microscopy image. Their experiments demonstrated the effectiveness of this virtual staining method on tissue sections such as human skin. This digital staining framework can further enhance the various uses of label-free QPI technology in pathology applications and general biomedical research, eliminating the need for histological staining, reducing costs and saving time associated with sample preparation. The implementation of a color opto-fluidic microscope SROFM prototype is reported. Based on a multi-wavelength illumination lens-less microscope, the system can scan approximately 400 cells per second for monochromatic imaging and 100 cells per second for color imaging, with the highest acuity optical resolution of 0.66 µm. They successfully applied the technique to color-image red blood cells infected with P. falciparum.

## 3. Smart-Phone Microscopy

The explosive application of smart phones not only provides a portable AI and tele- communication device, but also promotes a perfect and high-resolution camera. As shown in Figure 6, in the coming 5G telecommunication technology era, the development of modern mobile communication networks makes data transmission and remote information sharing. This also provides new development opportunities for computational, portable and low-cost microscopic imaging technology [72,73,74,75,76,77,78]. Thus, it is very potential to be applied to medical services in various remote and complex environments. The biological/medical images collected by smartphones can be transmitted to the central processing station for expert analysis or to a network server for cloud computing, such as image post-processing. Finally, the expert analysis results and image post-processing data are transmitted to the mobile phone for display, by the 5G+ networks and the Internet. Compared to the standard microscopic imaging systems, the smart phone’s back camera lens can be viewed as the tube lens and the CMOS image sensor, while an attached lens is the objective microscopic lens. According to the different attached lens, the microscopic imaging system based on the smartphone platform can be divided into three categories: the commercial microscopic objective lenses, the customized single lenses and the inverted pinhole lenses. As shown in Figure 7, there are three types of novel microscopic imaging optical path structures based on smartphone platforms. The first type of structure uses standard microscope objectives and eyepieces to form an infinity microscope optical system, and the smartphone platform is only used as an image recording and display device. Compared with traditional camera sensor acquisition equipment, the smartphone platforms, with integrated image acquisition and display features, provide strong compatibility, having the advantage of instant observation. The second type of structure uses the imaging lens and CMOS image sensor as an optical imaging and image acquisition device. Under this structure, a customized singlet lens with optical magnification capability is used as the objective lens for microscopic imaging, such as ball lens, aspheric lens or liquid lens. These customized singlet lenses are not sensitive to the gap between themselves with smartphone camera lens, so the total volume can be optimized. Distinct from the customized singlet lens design, since the multiple lens group matches the image sensor of the smartphone platform, the inverted pinhole/phone camera lens provides a third type of imaging optical path design, which can correct the image aberration and make better use of the image of the smartphone platform. As examples in Figure 8, based on the smartphone platform’s convenient and fast wireless data transmission capability, deep learning image enhancing and deep learning recognition algorithms would broaden applications of these cost-effective and portable microscopic imaging systems in medical diagnosis, disease diagnosis in remote areas testing and food safety testing [77,78].

### 3.1. Smartphone-Based Phase Contrast Microscopy

Image vision contrast in clinical microscopy is often achieved by chemical staining or labeling. By these dying methods, specific features of the sample can be enhanced, but they require extensive sample preparation. Label-free phase contrast imaging techniques do not require staining, and have not been developed for several years. Due to the higher cost of the associated optical hardware and the complexity of the system, their applications are limited. Now, computational, low-cost and portable microscopes based on smartphone platforms make label-free phase contrast techniques useful and possible in the fields of on-site testing and remote medical diagnosis [79,80,81,82]. To improve the quality of phase contrast images collected by smart-phone microscopes, deep learning networks can also transfer the style of directly collected images. Deep learning methods can extremely improve low-cost smartphone-based microscopes up to the imaging performance of bench-top microscopes. Bian et al. designed a portable microscope based on a smartphone platform, where an aspheric single lens is used as the imaging objective lens. The image data are transmitted to the computer through 5G/WIFI communication. Combined with the deep learning method of image style transfer, a virtual DPC image is obtained. Figure 9a,b show the structure and workflow of this setup. Traditional Zernike phase contrast microscopes require the insertion of a circular Zernike phase plate at the Fourier plane of the microscope objective. Also, traditional differential interference microscopes require finely assembled birefringent crystals such as Wollaston prisms. Now, only a self-designed aspheric single lens is used and a deep learning-based generative adversarial network to obtain a virtual phase contrast image with the same effect. Besides, the computational illumination could also achieve phase contrast microscopy, as shown in Figure 9c. These efforts by deep learning to transfer image style and improve the spatial resolution, will close the performance gap between smartphone microscopy and state-of-the-art bench-top DPC microscopy systems.

### 3.2. Smartphone-Based Dark-Field Microscopy

Similar to bench-top dark-field microscopes, in Figure 10, smartphone-based dark-field microscopes can also use ring illumination [83,84,85]. Instead of a dark-field condenser to achieve ring illumination, the outer ring of the LED array can be activated to provide ring illumination by using a programmable LED array. The LED array realizes dark field imaging by modulating the illumination aperture to be larger than the numerical aperture of the objective lens. Smartphone dark-field microscopy could also be based on total internal reflection (TIR) prism. TIR-based dark-field microscopy methods provide better signal-to-noise ratios than traditional oblique illumination-based dark-field microscopy, while eliminating the need for annular illumination apertures.

### 3.3. Smartphone-Based Quantitative Phase Microscopy

Digital holographic microscopes based on quantitative phase imaging have attracted much attention due to their quantitative phase imaging capabilities. Taking advantage of the cost-effective light source and compact system structure, various portable digital holographic microscopes have been designed based on smartphone platforms for sample observation and measurement [86,87,88]. Lee et al. also proposed a low-cost design for molecular diagnostics via digital holography on a smartphone platform [88]. Although LEDs can be used as light sources in portable smartphone digital holographic microscopes, a pinhole is still required to meet the requirement of illumination coherence. Furthermore, the high-resolution images would be reconstructed computationally based on diffraction algorithms. While both quantitative intensity and phase of the sample can be recovered from the hologram, the process is still time-consuming, including back-propagation and phase recovery. Based on these optical wave diffraction methods, Meng et al. developed an accurate, high-contrast, cost-effective, and portable phase imaging microscope based on a smartphone platform [86]. The system can be used to image biological samples such as Pap smears. In this TIE-based smartphone phase imaging microscope, the system resolution is ~1 μm, which can be used for 3D morphological studies of red blood cells. Phillips et al. used a hemispherical illuminator composed of an LED array as the illumination source and attached to an inverted smartphone platform-based microscope system. Compared to planar LED arrays, hemispherical LED arrays offer significantly better light efficiency, enabling shorter acquisition times and more efficient power usage.

### 3.4. Smartphone-Based Fluorescent Microscopy

When equipped with the appropriate accessories, fluorescence microscopy imaging is possible on a smartphone platform. This would provide more diagnostically-oriented biomedical observation [89,90,91,92,93,94,95,96,97,98,99,100,101,102,103,104]. A typical smartphone platform-based fluorescence microscope consists of an excitation light source (LED or laser diode), an emission filter, and an objective lens. In Figure 11a,b, smartphone fluorescence microscopes are proposed, which use an LD or an LED to excite the test tube sample. After passing through the sample, the fluorescence emission is collected perpendicular to the excitation direction. The key accessories of the epi-optical path are fluorescence excitation modules, which contain excitation filters, dichroic mirrors and emission filters. They can realize the orthogonality of the excitation optical path and the fluorescence emission optical path, and improve the signal-to-noise ratio of microscopy imaging. Wei et al. proposed a smartphone fluorescence microscope using laser diode oblique incidence illumination [100,103,104]. The sample is backlit by the excitation beam of a small laser diode with a wavelength of 450 nm at an incidence angle larger than the numerical aperture of the objective lens. This results in a high signal-to-noise ratio for imaging nano-scale analytes including DNA molecules, nanoparticles and viruses. Dai et al. used PDMS inkjet printing lens technology to design a lens with both focusing and filtering functions, and integrated the dual-function printing lens into a smartphone fluorescence microscope system [102]. The system’s structure is shown in Figure 11c. LEDs are used for bright-field imaging and LDs are used for fluorescence imaging. After inserting the LED chip or LD chip into the illumination source, the light source chip is positioned by two micro magnets and connected to the electrodes, which automatically turns on the LED or LD. The collimated laser beam irradiates the sample at an incident angle of 45°, which was greater than the acceptance angle of the dual function printing lens. Therefore, the excitation light is not directly coupled into the image sensor, effectively reducing the background noise.

### 3.5. Smart-Phone Microscopy Application Examples for POCT and Biomedicine Diagnosis

For sperm counting and monitoring, Computer Assisted Sperm Analysis (CASA) and Visual Assessment (VA) are the two assessment techniques used in the analysis. A method is proposed for sperm count analysis using smartphone microscopes and computers. [33] Smartphone microscopes are used to acquire image videos similar to visual assessment (VA) techniques; the videos of the samples are recorded by a high-resolution camera with a resolution of 1920 × 1080 and 30 Hz. Then the image data are wirelessly transmitted to the computer sever. A computerized sperm count software (CSCS) is designed to count and monitor sperm using a counting chamber. In contrast to the CASA system, a smartphone-based microscope offers a lower cost design. Compared with CASA and VA-based sperm count analysis, the proposed smartphone-based sperm concentration analysis is very attractive due to its modularity, functionality, accuracy, and cost. Using an LED light source, a dark-field condenser, and a 20× objective with a mobile phone camera, Sun et al. developed a dark-field smartphone microscope that can quantify nanoparticle signals for various research and medical applications [83]. The device captures images up to 8.0 megapixels. It weighs less than 400 g, and costs less than 2000 USDs. This method forms the basis of most clinical trials, combining kinetic and biomarker quantification, and provided a novel nanoparticle-based diagnostic assay for tuberculosis. The system is simple, robust, nanoparticle-based activity and quantitative analysis in resource-limited regions. Yang Zhenyu et al. demonstrate the integration of a quantitative phase imaging (QPI) method into a smartphone platform [87]. It is used for imaging red blood cells, with a resolution of about 1 μm. For algorithms, the computations actually run on a more powerful server. The computation time is less than 1 s. This device exhibits acceptable capabilities in erythrocyte imaging and reconstruction of cell thickness from computed phase maps for 3D morphological studies. In another example, a fluorescence microscopy imaging platform is developed on a mobile phone. [88] They demonstrated their platform for fluorescence imaging of labeled leukocytes in whole blood samples. Besides, water-borne pathogenic protozoan parasites, such as Giardia cysts, are successfully imaged over a large FOV of 81 mm^2^. A resolution of 10 µm is achieved. This compact and cost-effective fluorescence microscopy imaging platform weighs only about 28 g and measures about 3.5 × 5.5 × 2.4 cm. This setup could potentially be used for various lab-on-a-chip assays developed for global health applications, such as monitoring CD4-cell count or virus measurement for HIV patients.

## 4. Singlet Microscopy

Singlet lens is also another attractive way to achieve portable and low-cost microscopy setups. Commercial microscope objective lens and other imaging lens have become inexpensive due to mass industrial production. However, as they consist of multiple pieces of lenses, the cost mainly focuses on the lenses’ mounting and testing. In contrast, singlet lenses have no need for precise assembly, alignment and testing. The singlet lenses can reduce the time, money and labor cost extensively, resulting in a further price and integration revolution of imaging devices [105].

### 4.1. Singlet Bright-Field Microscopy

In Figure 12, the singlet microscopy setup is combined with only one aspheric lens and deep learning computational imaging technology [105]. The designed singlet aspheric lens is an approximate linear signal system. In this singlet microscopy setup, MTF curves on all FOVs are almost coincident with each other. The purpose of this design is to further improve imaging performance by using a deep learning algorithm. The total setup weighs only 400 g. By the sample of USAF-1951 target and pathological tissue slices, the experimental results show that both the resolution ability and FOVs of the singlet microscope are competitive with those of a commercial microscope with the 4X/NA0.1 objective lens. Figure 13 is the algorithm flowchart of the employed deep learning computational imaging method [105]. The algorithm includes two parts: the first is about the training stage of deep neural networks (DNN), and the other is the practical working stage. For the R/G/B illumination microscope, the data should be recorded separately, and the DNN training is executed respectively. The improved R/G/B channel images by deep learning would be combined as a colorful image computationally.

### 4.2. Singlet Achromatic Microscopy

The most attractive element of the singlet lens is freedom from precise testing, assembly, and alignment. These are very helpful for the application of portable and low-cost microscopes. But when the singlet is with only one kind of material, it would be difficult to overcome wavelength spectrum dispersion and chromatic aberrations. In Figure 14, the singlet lens and the deep learning image-style-transfer algorithms are combined to achieve achromatic aberrations [106]. These concepts and experiments have been proved and executed in clinical pathological slide microscopy. In the realm of hardware, the singlet aspheric lens is designed and fabricated. The lens has a high cutoff frequency and linear signal properties. There is only one mono-chromatic LED illumination, and the images are recorded by the CMOS image sensor. For algorithms, an image-style-transfer deep learning network is trained, which transfers mono-chromatic-illuminated greyscale microscopy images to virtual chemically stained images. A ‘U-Net’-like GAN framework architecture is designed to achieve image-style-transfer. Before the image-style-transfer, a conventional deep learning deconvolution method is interposed to improve the resolution and image contrast. In other computational art-applications, such as photo-to-comics, photo-to-painting, day-to-night and others, the textural details are not important. However, for medical and pathological observing, the image texture and high-resolution-content features should be kept in the virtually colorized images. In the proposed ‘U-NET’-like generator network, it strongly retains the high-resolution-content features of the original greyscale images. The loss function also contains two targets. One is to achieve the style transfer, and the other is to keep the high-resolution-content features. As shown in Figure 15, by experiments, data analysis and discussions, the proposed virtual colorization microscope imaging method is effective for H&E stained tumor tissue slides in singlet microscopy. It is believable that computational virtual colorization method for singlet microscopes would promote low-cost and portable singlet microscopy development in medical pathologic label staining observation (e.g., H&E staining, Gram staining, Fluorescent labeling, and so on).

### 4.3. Singlet Multi-Spectral Microscopy

To obtain more texture-spectrum information under different narrow wavelength spectrum bands, multi-spectral and hyper-spectral imaging methods and setups are proposed. Similarly, these portable and cost-effective multispectral microscopes are necessary in some remote biomedical applications. In Figure 16, the innovation concerns a portable and cost-effective multispectral microscopy setup [107]. In the optics hardware, a customized singlet lens is designed and fabricated. It could control rotational symmetric aberrations and eliminate the asymmetric optical aberrations. Then the image performance would be improved by the deep learning enhancing algorithms. The designed method helps to reduce the extreme difficulties of singlet lens fabrication due to the simple surface produced by the aberration optimization, while ensuring the high resolution. The singlet lens connects Zernike polynomial coefficients with singlet lens parameters through wavefront aberrations. By imaging a gold standard resolution pattern (Figure 17) and typical bio-samples, experiment results demonstrate that this portable singlet microscope would achieve multi-spectral microscopy well and cost-effectively.

### 4.4. Singlet Virtual Phase Contrast Microscopy

Phase imaging microscopy is for observing biological tissues and cells in vitro. Without chemical dying and fluorescent labeling, transparent and weakly scattering biological tissues/cells are imaged as the relative/quantitative phase information distribution. Conventional phase contrast microscopes consist of extensive precise and clean optics elements, which limits their usage, such as Zernike phase contrast (ZPC) microscopes (Figure 18a). In Figure 18, a singlet virtual Zernike phase contrast microscope is proposed for unstained pathological tumor tissue slide. In optics hardware, the objective lens is only one piece of lens. And there is no inset Zernike phase plate, which is even more expensive than a whole bright-field microscope setup. The Zernike phase contrast is virtually achieved by the deep learning computational imaging method. In the practical virtual Zernike phase contrast microscopy, the computational time consumed is less than 100 ms, which is far less than other computational quantitative phase imaging algorithms. By a conceptual demo experimental setup, it is competitive to a research-level conventional Zernike phase contrast microscope and effective for the unstained transparent pathological tumor tissue slides. Figure 18b is a singlet objective lens utilized to achieve virtual ZPC microscopy based on the deep learning computational imaging method [108]. The circular illumination part is still a Kohler illumination modulated by an annulus stop. However, instead of the conventional microscope objective consisting of multiple lens, a customized aspheric singlet lens is used here. The pathological tissue slide is at the objective plane, while a digital CMOS image sensor is at the conjugate imaging plane. But only based on the optic hardware in Figure 18b, one could not get a ZPC microscopy image of the transparent pathological tissue slice. One needs to computationally process the directly-recorded image by the deep learning ZPC-transfer method. In Figure 18c, the ‘using stage’ of the singlet virtual ZPC microscopy is presented. Before the ‘using stage’, ZPC-transfer DNN kernel would be deeply trained in the ‘training stage’. When the digital CMOS image sensor records an image, this image would be convoluted with the ZPC-transfer DNN kernel. Then the visual contrast of the microscopy image would be improved.

### 4.5. Singlet Meta-Lens Microscopy

Due to its ultrathin and flat structure, meta-lens has shown its wonderful capabilities in modulating and controlling light. Compared to the thick lens, the meta-lens has the most advantage of its high integration. This advantage would be very attractive for a portable microscope in the coming 5G+ telecommunication and AI era. Its high integration would be a good substitute for traditional thick lenses. In Figure 19, the ultrathin meta-lens is directly mounted on a CMOS image sensor, which constructs a highly integrated microscopy setup [109,110,111]. Different from conventional microscope objectives, the working distance is about sub-millimeters. The designed meta-lens is with NA0.78, made of GaN, in Figure 19a. The designed NA of this meta-lens objective lens is 0.37. The objective resolution is ~1.74 µm under the illumination wavelength of 630 nm, with a unit magnification. And the objective resolution is ~1.23 µm, with a 1.5× magnification. Besides, this meta-lens array approach would flexibly expand the FOV without sacrificing the resolution ability. This FOV extension method would achieve a high space-bandwidth product (SBP) for a wide-field microscopy application. Figure 19b shows the centimeter-scale FOV microscopy imaging, with a very high compact integration. But the illumination light should be polarized, with LCP light or RCP light. In this highly integrated microscopy setup, the meta-lens objective lenses are fabricated as a 6 × 6 meta-lens array. The total size of this setup is ~3.5 cm × 3 cm × 2.5 cm. Moreover, using the wavelength spectrum dispersion of the meta-lens, a highly integrated microscopy setup would develop tomography without mechanical scanning. Using the large diffractive dispersion property, this portable and cost-effective microscope would get an excellent tomographic imaging ability with a large DOF. And it successfully achieves the microscopic imaging for a frog egg cell. In conclusion, since the nature of ultrathin, flat and highly integration, the singlet meta-lens shows high potential in portable, low-cost and computational microscope in the 5G+ telecommunication and AI era.

### 4.6. Singlet Microscopy Application Examples for Biophotonics

Shen et al. propose a deep learning singlet microscope with imaging performance competitive with research-level commercial microscopes [81,105,106,107]. It has a total size of about 10 cm × 10 cm × 20 cm and weighs only 400 g. The resolution is up to 1.38 μm and a large FOV (diagonal 5 mm) is achieved. Since the singlet objective is a plastic model, it is very cost-effective compared to commercial objectives. The H&E stained pathological tumor tissue slides are successfully imaged multi-spectrally and colorfully. It can be seen that the portable singlet microscope has great application potential in biology, materials science, environmental science and other fields. Xu et al. demonstrate a compact imaging device with integrated meta-lens for wide-field microscopy [111]. The meta-lens is directly mounted on the image sensor. The device is based on a silicon lens working in the red wavelength range, with an overall size of about 3.5 × 3 × 2.5 cm. In this application, bio-samples of Pap smear and dragonfly wing are successfully imaged as the resolution of 1.74 um and a FOV of 1.2 × 1.2 mm^2^. Chen et al. proposed a metalens-based spectral imaging system that achieved high lateral and vertical resolutions, i.e., approximately 775 nm and 6.7 μm, respectively, with an aspheric GaN metalens (NA = 0.78) [110]. This computational portable microscopy setup successfully imaged the bio-sample of frog egg cells and showed excellent tomographic images of cell membranes and nuclei with distinct depth-of-focus (DOF) features.

## 5. Super-Resolution Microscopy

### 5.1. Super-Resolution Fourier Ptychography Microscopy

With optical microscopy developing, super resolution, large FOV, and phase imaging have been the hot researching focus. In some research-level microscopic setups, super-resolution and phase imaging capability have been outstanding advantages, compared with traditional microscopic imaging setups. But these advantages are at the cost of complex hardware structure, complicated operations and reducing other imaging performance, such as stochastic optical reconstruction microscopy (STROM), stimulated emission depletion (STED) microscopy and fluorescence emission difference (FED) microscopy. Fourier ptychographic microscopy (FPM) is an attractive and typical method to achieve QPI microscopy and super-resolution simultaneously, without affiliated precise optomechanical elements. Thus FPM has shown its high value and application prospect in low-cost portable microscopes [112,113,114,115,116]. In a typical FPM microscope, an LED array provides illumination from different directions. The CMOS image sensor collects a series of low-resolution images in turn. High spatial resolution, large FOV and quantitative phase imaging are achieved by computational phase recovery algorithms. Actually, FPM is the lens-based Fourier domain form of PIE, which is an extension of PIE phase recovery technology. The FPM uses a programmable LED array to provide flexible wavelength and illumination angles. FPM setups go beyond their super-resolution microscopy, and also computationally achieve quantitative phase imaging and tomography. Unlike a traditional Kohler lighting module, the illumination light source of FPM is a programmable LED array board. By sequentially illuminating each LED unit, the plane wave illuminates the bio-sample slide in different directions, while simultaneously the CMOS image sensor captures a series of low-resolution images. Each digitally recorded image corresponds to relative sub-spectral regions of the bio-sample. The illumination wave vector determines the center of the circular sub-spectral region. The illumination wavelength and the NA determine the radius of the region. In the assumed physical and mathematical model of FPM, the light wave from one LED, with wave vector of $\left(k_{x},k_{y}\right)$, illuminates the thin sample. This means a central shift $\left(k_{x},k_{y}\right)$ of the sample Fourier spectrum. The complex amplitude transmittance function t(x,y) can be used to represent the sample information. When the illumination plane wave goes through the sample, the distribution of the transmitted light can be expressed as: (3)$$E_{out}\left(x,y\right)=E_{in}\left(x,y\right)\cdot t\left(x,y\right),\;$$
where $E_{in}\left(x,y\right)$ is the complex function of the incident light. The FPM phase recovery process consists of the following five steps: 

(1)Generate the initial value of high-resolution complex amplitude in the spatial domain, $\sqrt{I_{h}}\exp\left(i\varphi_{h}\right)$.(2)Filter the illumination plane wave at a certain vector in the Fourier domain. And an inverse Fourier transform is implemented to generate a low-resolution image,$\;\sqrt{I_{1}}\exp\left(i\varphi_{1}\right)$.(3)The collected low-resolution image intensity $I_{lm}$ is replaced by $I_{1}$, and the corresponding sub-regions in the Fourier domain are updated.(4)Repeat Steps (2) and (3) for inclined plane wave irradiation from N vectors.(5)Repeat Step (2) to Step (4) for a new round of iterative update. The termination conditions for iteration updating can be set in advance.

According to Step (1) to Step (5) above, the inverse Fourier transform is performed in the Fourier domain, and then the high-resolution quantitative phase information is obtained in the spatial domain. Based on this FPM framework, some low-cost computational super-resolution microscopes have been achieved.

### 5.2. Super-Resolution Lens-Less Microscopy Based on Sub-Pixel Displacement

The sub-pixel displacement of the sampling plane with respect to the CMOS image sensor is one of the super-resolution methods [117,118,119,120]. This mechanical lateral displacement could be carried out by displacement of the sample/sensor or moving the light source. When the sample/sensor is being displaced, the displacement accuracy must be at the sub-pixel level. This is usually the sub-micron displacement, which will increase the cost significantly. However, using the lateral displacement of the light source to obtain sub-pixel shifted images can greatly reduce the cost. Assuming that the gap between the light source and the sample is z1, the gap between the sample and the CMOS image sensor is z2. In a unit-magnification lens-less microscope, z1 is much longer than z2, i.e., z1 >> z2. The ratio between the light source’s lateral movement and the corresponding displacement of the CMOS image sensor is z1/z2. Therefore, moving the light source would reduce the precision requirement of the system’s mechanical displacement. As a consequence, this method of improving the resolution achieved by the mechanical displacement of the light source is widely used in lens-less super-resolution phase imaging. Although moving the light source has lower requirements on the accuracy of the stage than the displacement sensor/sample, mechanical displacement is still required. In order to reduce this mechanical displacement, the light is coupled to different fibers. After optical fiber coupling, these fibers are accurately assembled into a small area, which not only realizes the light source’s lateral displacement, but also promotes the spatial coherence of the light source.

### 5.3. Super-Resolution Lens-Less Microscopy Based on Wavelength Scanning

Similar to scanning the angles and sub-pixel, wavelength scanning would also achieve super-resolution lens-less microscopy. A wavelength-tunable laser would be used to generate wavelengths ranging from 498 nm to 510 nm (with an interval of 3 nm), and each dominant wavelength corresponds to a spectral width of about 2 nm [121]. Furthermore, additional multi-angle illumination and multi-height were added during the experiment. For the algorithm, a high-resolution correction matrix process for calculating limited light intensity is incorporated into the phase recovery method based on synthesizing apertures. Reference [119] shows the schematic diagram of the device. Based on 60 holograms, 5 angles of illumination and 12 wavelengths of illumination, the pixel size of the final 1.12 μm camera achieves a resolution of 0.5 μm.

### 5.4. Super-Resolution Lens-Less Microscopy Based on Multi-Angle Illumination

In super-resolution lens-less microscopes based on multi-angle illumination (Figure 20), the light source is mounted on a rotating arm in order to achieve oblique illumination of two orthogonal axes. In the super-resolution algorithm, the light source has a lateral motion at every angle. Thus, the high-resolution hologram images are generated by synthesizing holograms at each angle [122]. This approach is extremely similar to the FPM framework. Different angles of light are equal to the synthesis of Fourier spectrum information. The reconstruction relays on the derived G-S recovery algorithms backward and forward, between the spatial domain and the Fourier domain. By recovering the super-resolution complex amplitude information at the sensor plane, the information is propagated back to the object surface through numerical diffraction calculation. Ultimately, the objective light intensity and phase distribution with high resolution are obtained. Actually, sub-pixel displacement, wavelength scanning and multi-angle illumination could be combined simultaneously to achieve lens-less super-resolution.

### 5.5. Super-Resolution Lens-Less Microscopy Based on Axial Scanning

Axially scanning the distance is used between the CMOS image sensor and the sample for super-resolution lens-less microscopy [123]. The CMOS image sensor collects low-resolution light intensity in different defocus planes. After computational reconstruction, pixel super-resolution and phase recovery would simultaneously be achieved. This method introduces pixel transfer function into the traditional iterative phase recovery method. TIE is used to solve the phase as the initial value of iterative reconstruction, which can significantly accelerate the speed of iterative reconstruction. This proposed adaptive G-S iterative phase recovery imaging algorithm can be considered as an incremental gradient descent optimization algorithm. This algorithm not only maintains the fast convergence of the initial iteration by the incremental gradient method, but also remarkably improves the robustness. With the camera pixel size of 1.67 μm, the half-width resolution of 0.77 μm was finally achieved based on 10 defocus holograms.

### 5.6. Portable Super-Resolution Microscope Applications on Biomedical Observation

For observing CTCs, using a portable FPM setup, the researchers obtained high resolution color images of large FOV micro-filter samples, with a quantitative phase data. This portable FPM setup can refocus the 300 μm depth range of the sample. The sample image acquisition time is about 3 min, the reconstruction time of each color channel is about 10 min, and it takes a total of 39 min to generate a color image of a large FOV with a quantitative phase data. This portable FPM setup demonstrated high image quality, efficiency, and consistency in detecting tumor cells when the corresponding micro-filter samples were compared to standard microscopes with high correlation (R = 0.99932). Bishara et al. report a sub-pixel-shift-based lens-less super-resolution microscope. It is a compact on-chip microscope weighing approximately 95 g [71]. It is capable of reconstructing holographic images (amplitude and phase) of observed objects (e.g., human malaria parasites) with a resolution of less than 1 µm. Its FOV reaches up to 24 mm^2^. This lens-less on-chip microscope has successfully imaged malaria parasites. The results show that a compact and lightweight on-chip microscope is important for addressing global health problems such as diagnosing infectious diseases in remote areas. Luo et al. report a wavelength-scanning-based pixel super-resolution technique [121]. The technique allows analysis of achromatic (e.g., unstained) and stained/stained organisms. The red blood cells with a large area is used as the typical bio-sample. Besides, Luo et al. also provide a multi-angle illumination lens-less super-resolution microscopy technique [122]. To demonstrate the validity of this synthetic aperture-based holographic on-chip microscope, unstained Pap smears are successfully imaged on a very large FOV of 20 mm^2^. This compact synthetic aperture-based approach on-chip microscopes can be used in a variety of applications in medicine, physical science, and engineering that require high-resolution, wide-field imaging.

## 6. Discussion

The above computational portable microscopes show attractive potential applications for point-of-care-tests and tele-diagnosis. But, as developers and users, we should not be too optimistic. Usually, a computational portable microscope is developed for certain biomedical applications. Once designed, its function is almost unchangeable. That means one setup for one application. When faced with other applications, the setup should be modified slightly. Besides, for computational algorithms, unfortunately, some popular computational approaches are also marred by lack of universality, risking the generation of errorri artifacts. For example, the well-known issues with ‘over trained’ AI-based algorithms. It is not a big problem to generate a nearly ideal AI model for the selected data set, but design and validation of the robust, stable, and universal AI-algorithms is a different story. As for validation, some proof-of-concept studies do not cover this part much. Thirdly, low-cost is another claimed advantage. Commonly, research-level and desktop microscopes have been widely used in the medical centers and scientific institutes, which are relatively expensive. For example, a bright-field microscope is about $3000, a dark-field microscope is about $3000, a fluorescence microscope is about $10,000, a Zernike-phase-contrast (ZPC) microscope is about $20,000, a differential-interference-contrast (DIC) microscope is about $20,000, a laser confocal microscope is about $0.15 million and a super-resolution microscopy imaging system is about $1 million. Although computational portable microscopes are not widely purchased as a universal commercial price, the cost of computational portable microscope would be estimated as Table 1.

## 7. Conclusions

With the needs of POCT and tele-diagnosis increasing, computational portable microscopes become more and more necessary in bio-photonics and bio-medicine applications. As electronic products, home computers, computer sever stations and AI algorithms develop fast. Their costs are approaching an acceptable level due to the industrial mass scale effect. Thus, these cost-effective optical, electronic and computational resources are easy to get [124]. Obviously, in above mentioned lens-less microscopy setups, smart cellphone microscopes, singlet lens microscopes and super-resolution portable microscopes, the components are easy to be collected by a designer. Moreover, the widely applied 5G+ telecommunication and WIFI wireless networks make it very convenient to transfer big-data, for example, high resolution images. The available computer sever station and cloud storage space would provide cheap computational resources, with a powerful hash-rate. These available hardware and software environments provide the base condition for developing computational, portable and low-cost microscopes. Briefly, this is the best era for POCT and tele-diagnosis [124,125,126,127]. It is believable that more ideas and innovations for developing computational, portable and low-cost microscopes for POCT and tele-diagnosis would boom.

## Figures and Tables

**Figure 1 cells-11-03670-f001:**
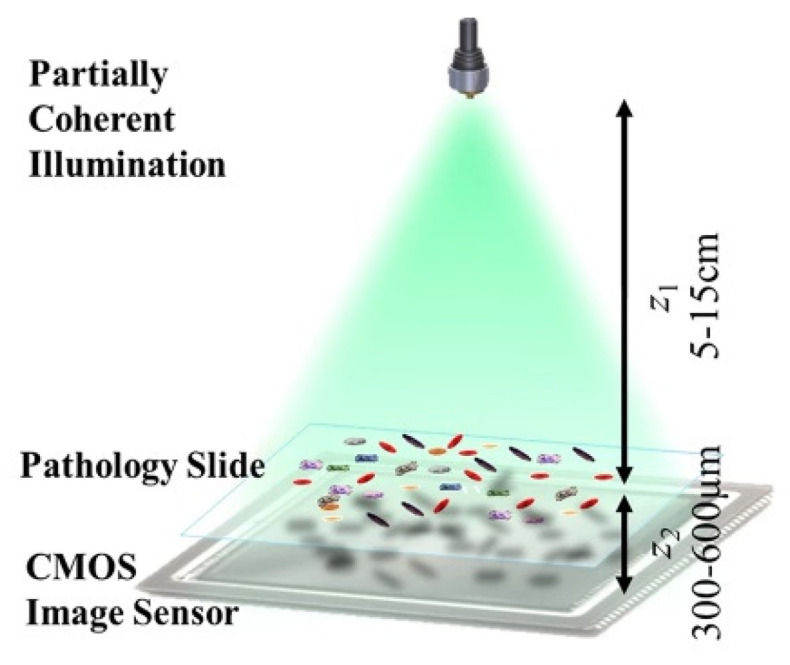
Typical schematic of lens-less diffraction microscopy setups based on a single frame [39].

**Figure 2 cells-11-03670-f002:**
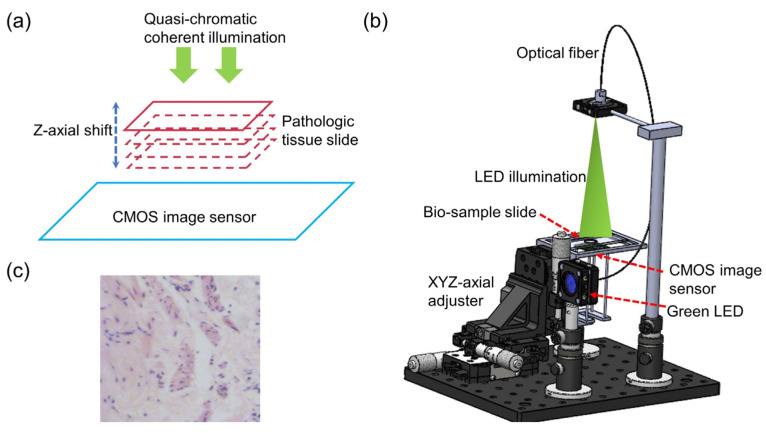
Typical schematic of lens-less diffraction microscopy setups based on multiple holograms: (**a**) the principal schematic; (**b**) a 3D mechanical design of a lens-less diffraction microscope; (**c**) typical H&E stained pathological slide [57].

**Figure 3 cells-11-03670-f003:**
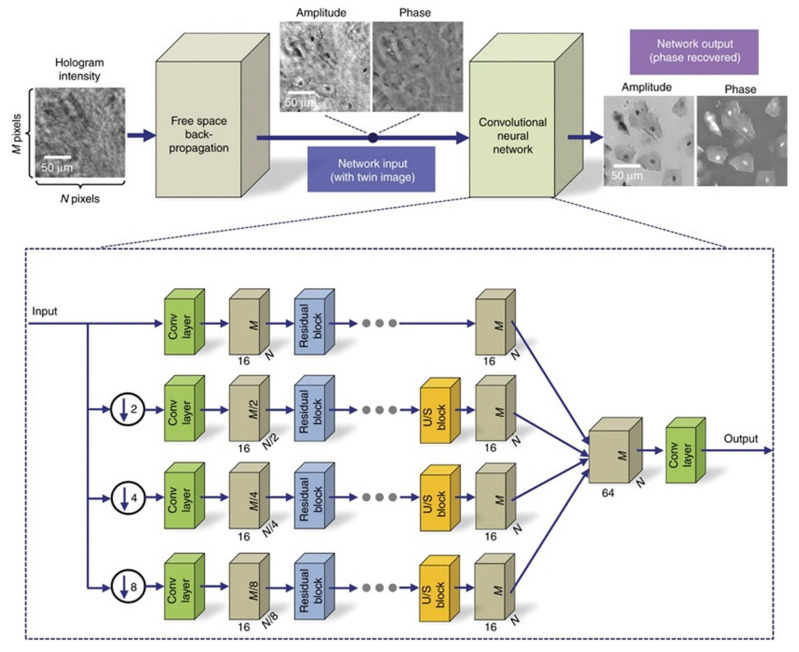
CNN architecture to recover complex-value information of the object [59].

**Figure 4 cells-11-03670-f004:**
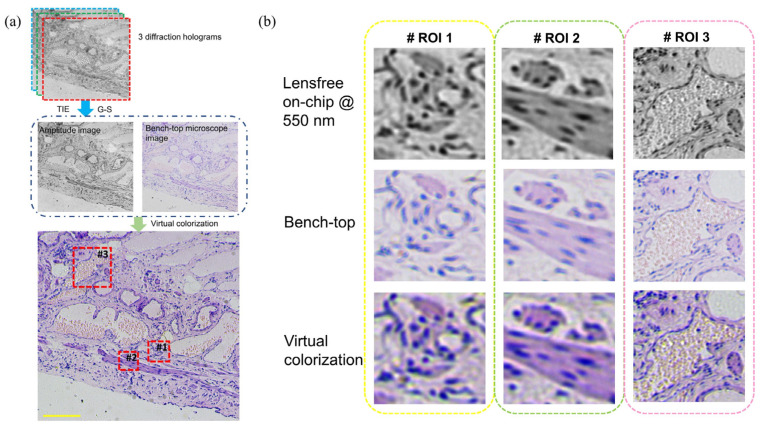
(**a**,**b**) Deep learning virtually colorizing to achieve colorful lens-less microscopy[57].

**Figure 5 cells-11-03670-f005:**
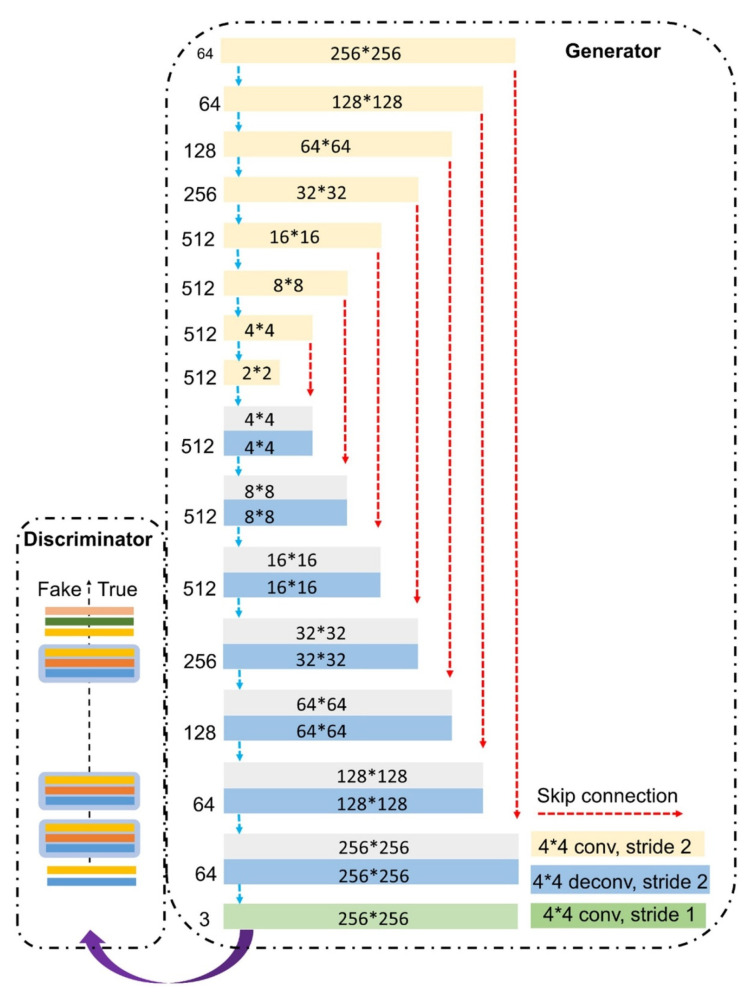
Deep learning virtually colorizing based on a GAN architecture [57].

**Figure 6 cells-11-03670-f006:**
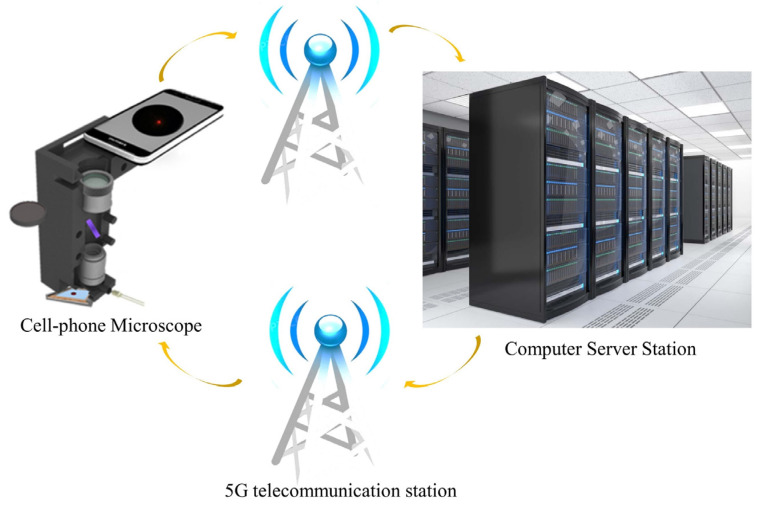
Interaction pipeline maps of cell-phone microscopes, 5G telecommunication stations and computer sever stations.

**Figure 7 cells-11-03670-f007:**
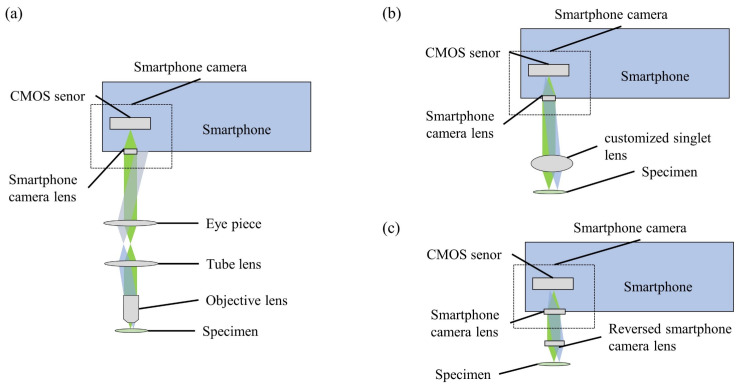
Smartphone platforms based on: (**a**) commercial microscopic objective lenses, (**b**) customized single lenses, (**c**) inverted pinhole lenses.

**Figure 8 cells-11-03670-f008:**
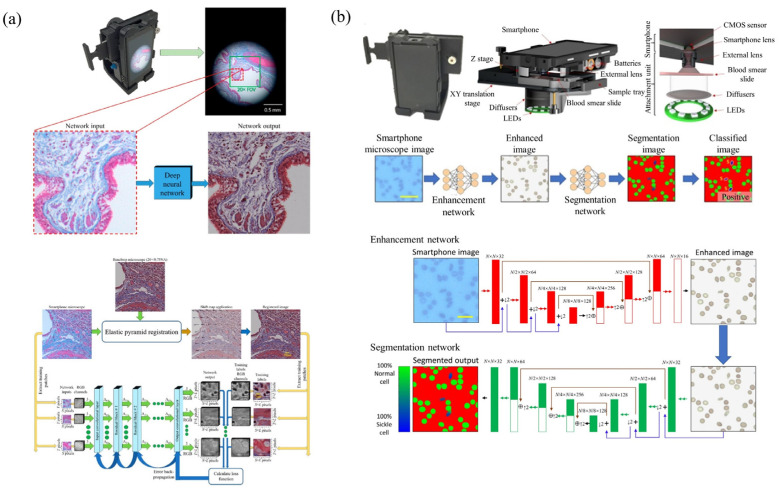
Smartphone-based bright-field microscope with deep learning. (**a**) Deep learning enhanced mobile-phone microscopy [77], and (**b**) a photograph of the smartphone-based bright-field microscope. [78].

**Figure 9 cells-11-03670-f009:**
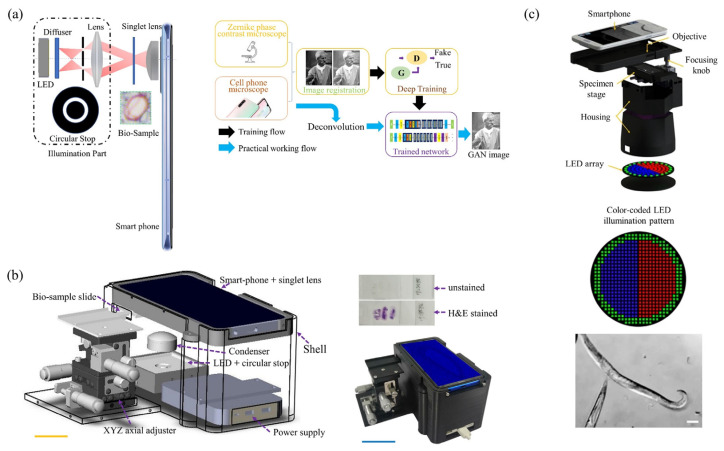
Smartphone-based phase contrast microscopy setups: (**a**) a 2D principle schematic based on a singlet objective and deep learning architecture for image enhancing [81]; (**b**) a 3D mechanical design [81]; (**c**) smartphone-based phase contrast microscope based on computational LED illumination [79].

**Figure 10 cells-11-03670-f010:**
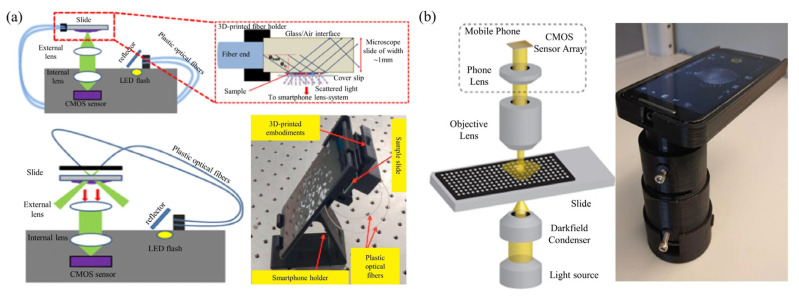
Smartphone-based dark-field microscopy based on (**a**) external angular illumination [84], and (**b**) a dark-field condenser [83].

**Figure 11 cells-11-03670-f011:**
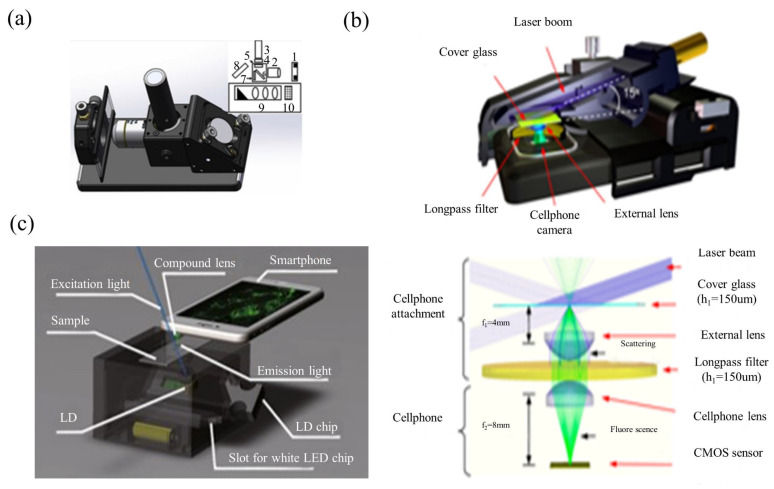
Smartphone-based fluorescent microscopy setups. (**a**) schematic illustration of the smartphone microscope [90]; (**b**) photograph and schematic of cell-phone-based fluorescence microscopes. [104]; and (**c**) schematic diagram of portable fluorescence microscopes [102].

**Figure 12 cells-11-03670-f012:**
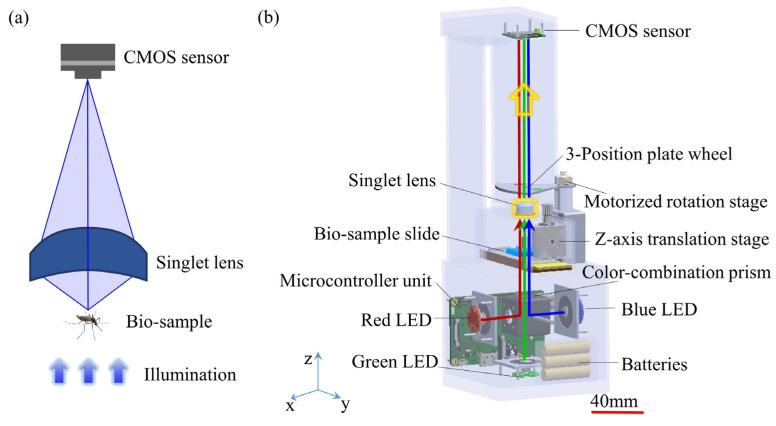
Singlet bright-field microscopy setup [105]: (**a**) a 2D principle schematic and (**b**) a 3D mechanical design.

**Figure 13 cells-11-03670-f013:**
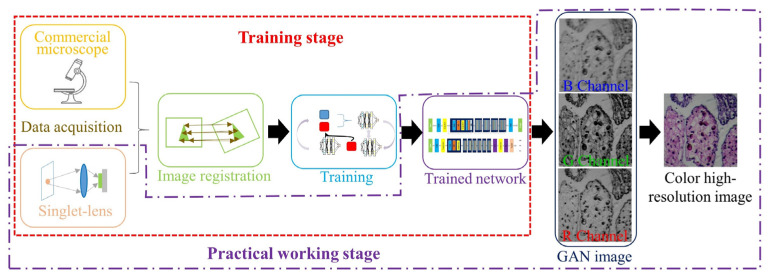
Deep learning architecture to enhance high-resolution and image contrast for the singlet bright-field microscopy in Figure 12 [105].

**Figure 14 cells-11-03670-f014:**
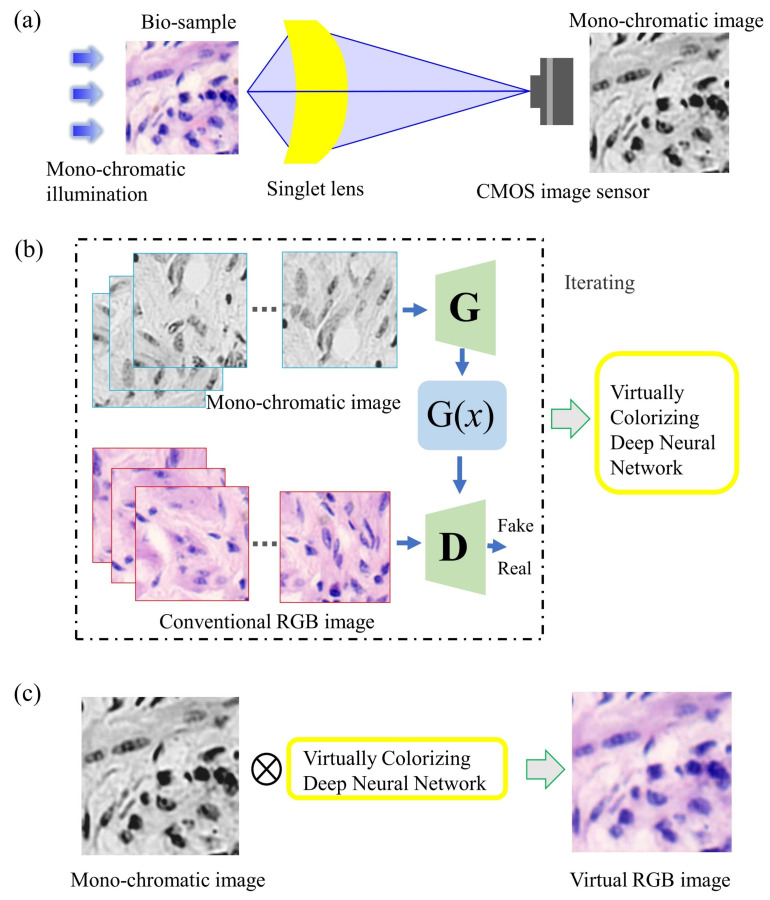
Singlet achromatic microscopy based on deep learning virtual colorization [106]. (**a**) 2D schematic; (**b**) a deep learning virtualizing flowchart; and (**c**) an application working flowchart.

**Figure 15 cells-11-03670-f015:**
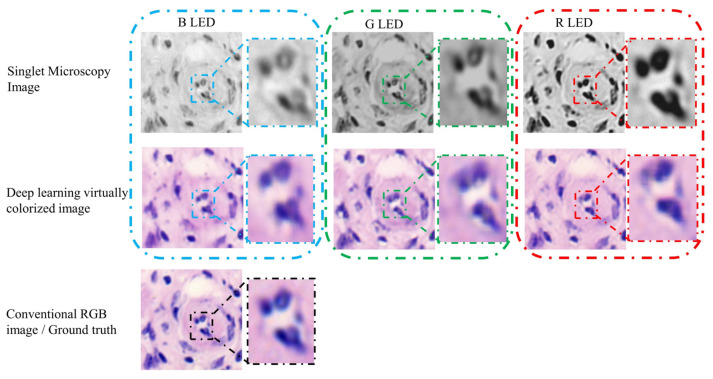
Experimental singlet achromatic microscopy results [106].

**Figure 16 cells-11-03670-f016:**
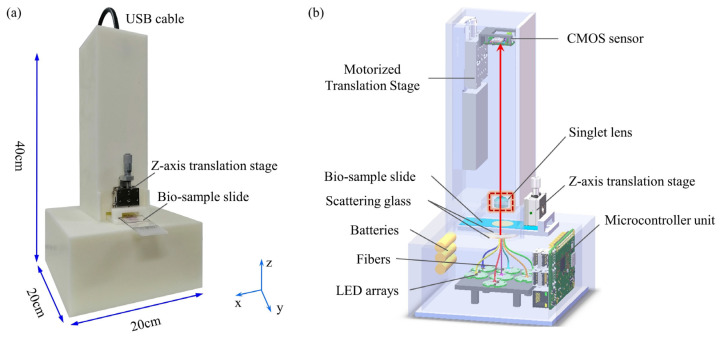
Singlet multi-spectral microscopy [107]: (**a**) photograph, and (**b**) a 3D mechanical design.

**Figure 17 cells-11-03670-f017:**
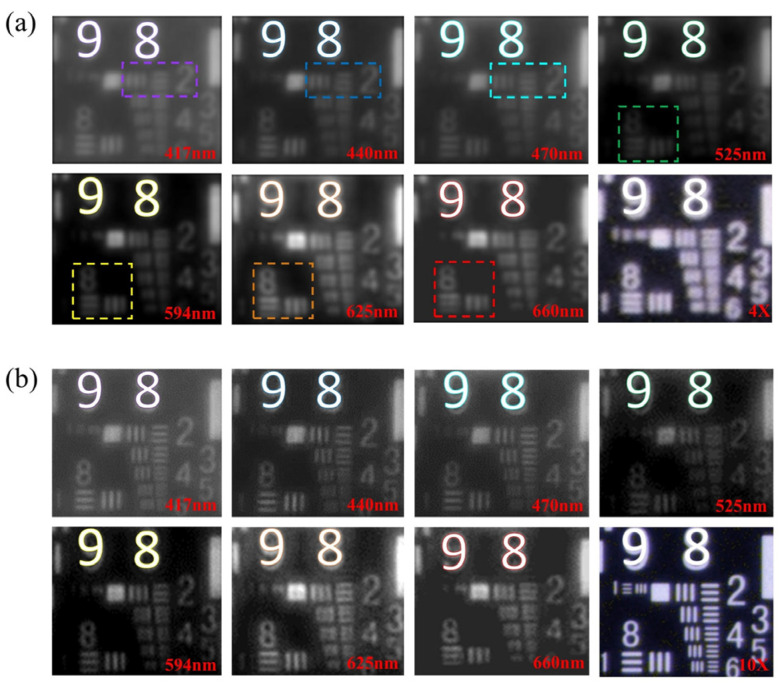
Experimental singlet multi-spectral microscopy by a sample of USAF-target resolution plate [107]. (**a**) Sub-picture 1–7 immediately imaged by singlet MSM with 7 wavelength, and sub-picture 8 was imaged by commercial microscope with 4X objective. (**b**) Sub-picture 1–7 recovered by deep learning algorithm with 7 wavelength, respectively; and sub-picture 8 was imaged by commercial microscope with 10X objective.

**Figure 18 cells-11-03670-f018:**
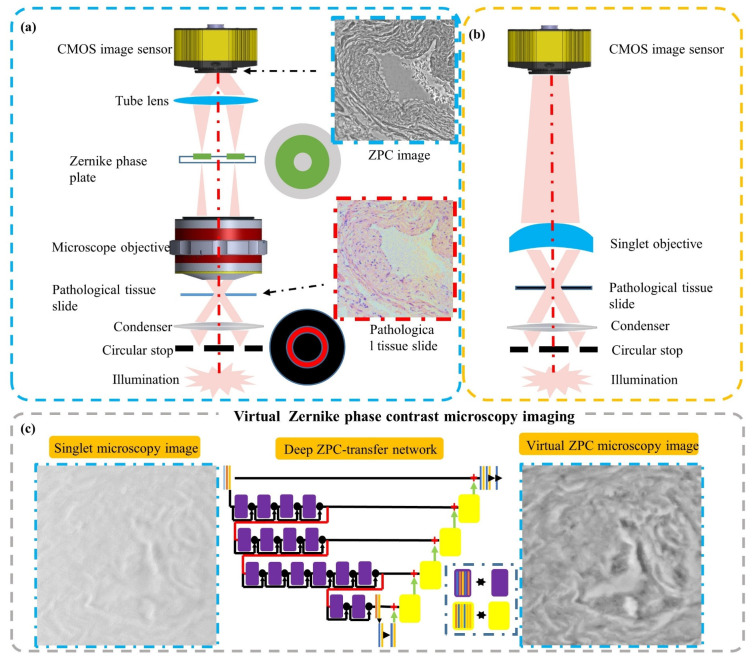
Singlet virtual phase-contrast microscopy. [108]: (**a**) a 2D schematic of a traditional Zernike phase contrast microscopy setup, (**b**) a 2D schematic of a singlet phase contrast microscopy setup, and (**c**) Deep ZPC-transfer network architecture.

**Figure 19 cells-11-03670-f019:**
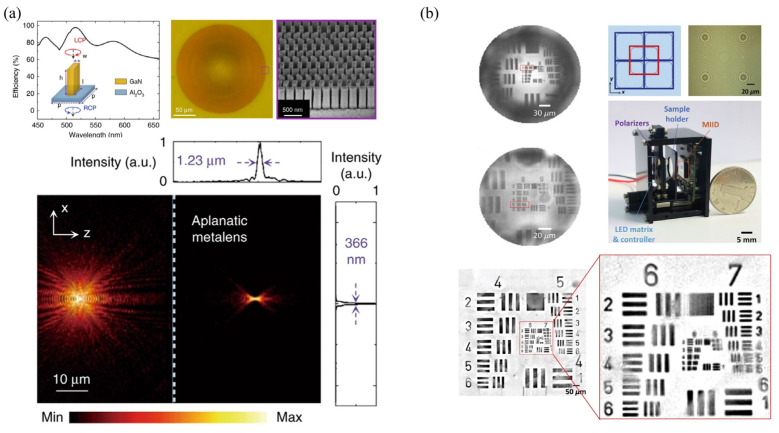
Portable singlet meta-lens microscopy setup: (**a**) macro-scale and nano-scale structure of a singlet meta-lens [110] and (**b**) a photograph of the singlet meta-lens microscopy setup with the resolution test result [111].

**Figure 20 cells-11-03670-f020:**
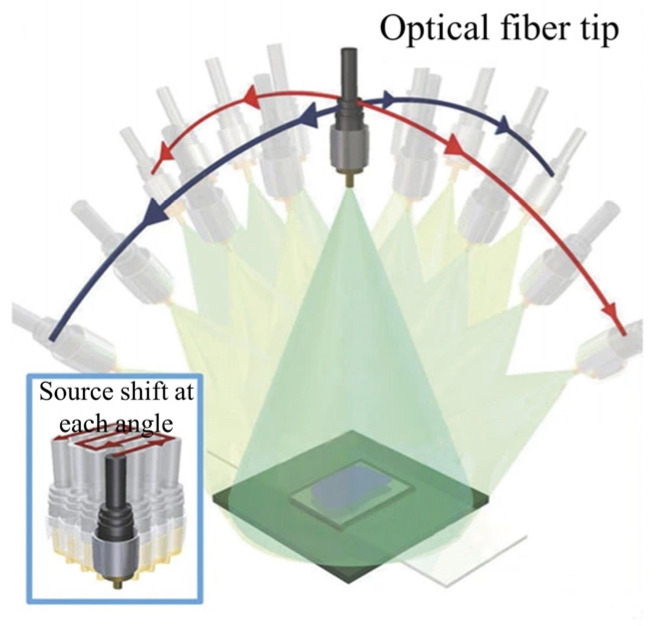
Super-resolution lens-less microscopy based on multi-angle scanning [122].

**Table 1 cells-11-03670-t001:** Estimated prices of key optical and electronic elements in computational portable microscopes.

Type	Elements	Cost
Lens-less Microscopy	Light source	under $1
	Sensor	$50 to $200
	Structure	under $10
Smart-phone microscopy	light source	under $1
	Smartphone	under $800
	Objective lens	$10 to $100
	Customized singlet lens	under $100
	Reversed smartphone camera lens	under $50
	structure	under $10
Singlet microscopy	light source	under $1
	Sensor	$50 to $200
	Singlet lens	under $100
	structure	under $10
Super-resolution microscopy	light source	~$500
	Sensor	$50 to $200
	Lens	$10 to $100
	structure	under $100

## Data Availability

The data that support the findings of this study are available from the corresponding author upon reasonable request.

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
