# Peer review of "Computational Portable Microscopes for Point-of-Care-Test and Tele-Diagnosis"

_cells, 2022, doi:10.3390/cells11223670_

Round 1

Reviewer 1 Report

This manuscript, presented by Yinxu Bian et al., reviews different computational, portable, and low-cost microscope devices used as point-of-care tests (POCT) and telediagnosis. I think it is well-written and technically sound; it is easy to read and comprehensive. I believe this manuscript would interest those involved in implementing novel and low-cost remote microscope techniques; however, I suggest minor revisions before being accepted for publication.

Minor comments and suggestions:

The main drawback of this manuscript is the lack of explanation of many figures. They have too much information not explained in the text or figure captions, containing incises that are not described. A brief description in the figure caption is strongly recommended and should be referenced if the figure is taken from a particular citation; so that the reader can find more specific information.

On page 9, line 327, a figure depicting the three types of microscopic imaging optical path structures mentioned would be helpful to understand better the main differences described in the text.

Figures 1, 12, and 13 are not referenced in the text.

Finally, I suggest citing the previous review of Hussain I and  Bowden AK. “ Smartphone-based optical spectroscopic platforms for biomedical applications: a review [Invited] .”Biomed Opt Express. 2021; 12(4):1974-1998. DOI: 10.1364/BOE.416753. PMID: 33996211; PMCID: PMC8086480. Who summarized the development of different smartphone-based spectroscopic systems by highlighting the challenges and potential solutions in achieving affordability, portability, higher accuracy, and adaptability for point-of-care applications. Possibly stating the main difference between this manuscript and the review mentioned above in the introduction section would be enough.

Author Response

Response to Reviewer 1

--------------------------------------------------------------------------------

Q1: A brief description in the figure caption is strongly recommended and should be referenced if the figure is taken from a particular citation; so that the reader can find more specific information.

Authors’ response: Thanks. Long figure captions have been added. And the references are provided.

Q2: On page 9, line 327, a figure depicting the three types of microscopic imaging optical path structures mentioned would be helpful to understand better the main differences described in the text.

Authors’ response: Thanks. Optical path structures are added.

Q3: Figures 1, 12, and 13 are not referenced in the text.

Authors’ response: Thanks. We have corrected.

Q4: Finally, I suggest citing the previous review of Hussain I and Bowden AK. “Smartphone-based optical spectroscopic platforms for biomedical applications: a review [Invited].”Biomed Opt Express. 2021; 12(4):1974-1998. DOI: 10.1364/BOE.416753. PMID: 33996211; PMCID: PMC8086480. Who summarized the development of different smartphone-based spectroscopic systems by highlighting the challenges and potential solutions in achieving affordability, portability, higher accuracy, and adaptability for point-of-care applications. Possibly stating the main difference between this manuscript and the review mentioned above in the introduction section would be enough.

Authors’ response: Thanks. We have added in in the introduction section.

Reviewer 2 Report

The review is comprehensive, deep, and broad. The timing for this review is perfect. Many new technological solutions proposed within the last decade and especially the last few years make advance imaging more accessible. This review satisfies the ever-growing demand for a systematized “bibliography” of the recent advances. The overall impression is very positive. However, there are several moments to consider here. The very general comment – this review is a little bit too optimistic. While positive moments and trends were very well covered, some problems were not fully addressed . Many reviewed methods are heavily based on mathematical algorithms.  Unfortunately, many popular computational approaches can generate artifacts.  For example,  the well known issues with “over trained “ AI-based algorithms. It is not a big problem to generate a nearly ideal AI model for the selected data set, but design and validation of the robust, stable, and effective AI-algorithms is a different story. As for validation, some proof-of-concept studies do not cover this part much. Some, critics, and comments on this matter would be very useful.  Another moment, the “older” microscopy methods accompanied by older references mainly (see comments below). It is misleading. As we know, scientist did not stop their efforts to further improve these methods. It should be acknowledged by references and comments. Some references are not quite consistent with the local context. Please, check reference for duplicates. I spotted several. With this number of images, it would be nice to have references within every figure caption.

This review emphasizes the lower cost of the discussed imaging systems. If authors did aggregate this kind of information, why would not combine known real or estimated costs of the new and conventional systems in one table? Many readers would be happy to see these numbers

Any other recent reviews covering the same matter?

Could you add “Contents pages”? It is not easy to navigate through many sections with similar titles.

Additional comments:

Line 58. “POCT diagnostic microscopy devices generally meet…”  - it is too optimistic. Users do want to have POCT device fast, simple, accurate, and small, right, but very often these devices do not meet all characteristics. 

Line 57. Ref 17 – I am not sure why this reference was used here. 

Line 57. Ref 18 - BZ-8000 microscope is a relatively small table-top microscope but it is not very portable.

Line 71. Ref 24 – it is a news article. The original paper should be referenced. It is not clear why ref #24 was used in connection with the 5G+ telecommunication and AI-era

Line 90, ref 27-34 – the newest publication is 2014. There are many interesting examples published within the last 5 years.

Line 116, ref 35-39 – the newest publication is 2014. There are newer articles to mention

Line 144, ref 42 – this paper more about fundamental studies rather than about a real-life application of the proposed technology

Line 162. ref 43-45 – no significant publications after 2017? Two old articles on basic principles and one 2017 technical/application paper do not represent the current stage of the “Phase Recovery Based on Single Frame” area.

Line 277 – reference 36 is not about C. elegans

Line 296 – why authors decided to use Rivenson, Rivenson is not among three main authors who contributed equally

Line 354  “… be enhanced, by…” – Is it “but”?

Ref 75 – should it be 2018?

Ref 72 and ref 76 – the same paper

Ref 73 and ref 82 - the same paper

Line 451 – I am not sure that ref 65 and 69 can be referenced in the context of “sperm count analysis”

Line 460 – Reader should find which reference (77 or 104) was referenced here.

Line 503 – please add reference/s

Line 795 – “computer sever station” – server?

Line 791 – I am not sure that the ref #116 is a good selection for the context.

Author Response

Response to Reviewer 2

--------------------------------------------------------------------------------

Q1: This review is a little bit too optimistic. While positive moments and trends were very well covered, some problems were not fully addressed. Many reviewed methods are heavily based on mathematical algorithms. Unfortunately, many popular computational approaches can generate artifacts.  For example, the well known issues with ‘over trained’ AI-based algorithms. It is not a big problem to generate a nearly ideal AI model for the selected data set, but design and validation of the robust, stable, and effective AI-algorithms is a different story. As for validation, some proof-of-concept studies do not cover this part much. Some, critics, and comments on this matter would be very useful.

Authors’ response: It is a good and critical suggestion. We have add these critics and comments in the section ‘Discussion’, about computational algorithm drawbacks.

Q2: Another moment, the “older” microscopy methods accompanied by older references mainly (see comments below). It is misleading. As we know, scientist did not stop their efforts to further improve these methods. It should be acknowledged by references and comments. Some references are not quite consistent with the local context. Please, check reference for duplicates.

Author response: Thanks. We have checked and corrected.

Q3: This review emphasizes the lower cost of the discussed imaging systems. If authors did aggregate this kind of information, why would not combine known real or estimated costs of the new and conventional systems in one table? Many readers would be happy to see these numbers. Any other recent reviews covering the same matter?

Authors’ response: Thanks. Discussion and comparison data table are in the new section ‘Discussion’.

Q4: Could you add ‘Contents pages’? It is not easy to navigate through many sections with similar titles.

Authors’ response: Thanks. ‘Contents page’ will be discussed with the journal office, as the page numbers should be consistent with the journal’s.

Q5: Line 58. “POCT diagnostic microscopy devices generally meet…” - it is too optimistic. Users do want to have POCT device fast, simple, accurate, and small, right, but very often these devices do not meet all characteristics.

Authors’ response: Thanks. The statement has been optimized.

Q6: Line 57. Ref 17 – I am not sure why this reference was used here.

Authors’ response: Thanks. We have checked and updated.

Q7: Line 57. Ref 18 - BZ-8000 microscope is a relatively small table-top microscope but it is not very portable.

Authors’ response: Thanks. We have checked and updated.

Q8: Line 71. Ref 24 – it is a news article. The original paper should be referenced. It is not clear why ref #24 was used in connection with the 5G+ telecommunication and AI-era.

Authors’ response: Thanks. We have checked and updated.

Q9: Line 90, ref 27-34 – the newest publication is 2014. There are many interesting examples published within the last 5 years.

Authors’ response: Thanks. We have checked and updated.

Q10: Line 116, ref 35-39 – the newest publication is 2014. There are newer articles to mention.

Authors’ response: Thanks. We have checked and updated.

Q11: Line 144, ref 42 – this paper more about fundamental studies rather than about a real-life application of the proposed technology.

Authors’ response: Thanks. We have checked and updated.

Q12: Line 162. ref 43-45 – no significant publications after 2017? Two old articles on basic principles and one 2017 technical/application paper do not represent the current stage of the “Phase Recovery Based on Single Frame” area.

Authors’ response: Thanks. We have checked and updated.

Q13: Line 277 – reference 36 is not about C. elegans.

Authors’ response: Thanks. We have checked and updated.

Q14: Line 296 – why authors decided to use Rivenson, Rivenson is not among three main authors who contributed equally.

Authors’ response: Thanks. We have checked and updated.

Q15: Line 354 “… be enhanced, by…” – Is it “but”?

Authors’ response: Thanks. We have checked and updated.

Q16: Ref 75 – should it be 2018? Ref 72 and ref 76 – the same paper. Ref 73 and ref 82 - the same paper.

Authors’ response: Thanks. We have checked and updated.

Q17: Line 451 – I am not sure that ref 65 and 69 can be referenced in the context of “sperm count analysis”.

Authors’ response: Thanks. We have checked and updated.

Q18: Line 460 – Reader should find which reference (77 or 104) was referenced here.

Authors’ response: Thanks. We have checked and updated.

Q19: Line 503 – please add reference/s.

Authors’ response: Thanks. We have checked and updated.

Q20: Line 795 – “computer sever station” – server?

Authors’ response: Thanks. We have checked and updated.

Q21: Line 791 – I am not sure that the ref #116 is a good selection for the context.

Authors’ response: Thanks. We have checked and updated.
